# Modulation of Chitosan-TPP Nanoparticle Properties for Plasmid DNA Vaccines Delivery

**DOI:** 10.3390/polym14071443

**Published:** 2022-04-01

**Authors:** Renato Nunes, Ana Sofia Serra, Aiva Simaite, Ângela Sousa

**Affiliations:** 1CICS-UBI—Health Sciences Research Center, University of Beira Interior, Avenida Infante D. Henrique, 6200-506 Covilhã, Portugal; renato.nunes@ubi.pt (R.N.); anasofiamms@gmail.com (A.S.S.); 2InoCure s.r.o, R&D Laboratory Center, Prumyslová 1960, 250 88 Celákovice, Czech Republic; aiva.simaite@gmail.com

**Keywords:** cervical cancer, chitosan-TPP nanoparticles, ionotropic gelation, plasmid DNA vaccine

## Abstract

Nucleic acid vaccines have become a revolutionary technology to give a fast, safe, cost-effective and efficient response against viral infections, such as SARS-CoV-2 or Human papillomavirus (HPV). However, to ensure their effectiveness, the development of adequate methods to protect, carry, and deliver nucleic acids is fundamental. In this work, nanoparticles (NPs) of chitosan (CS)-tripolyphosphate (TPP)-plasmid DNA (pDNA) were thoroughly modulated and characterized, by measuring the charge and size through dynamic light scattering (DLS) and morphology by scanning electron microscopy (SEM). Stability, cytotoxicity and cellular uptake of NPs were also evaluated. Finally, the effect of polyplexes on the expression of HPV E7 antigen in human fibroblast and RAW cells was investigated through polymerase chain reaction (PCR) and real-time PCR. The results showed NPs with a spherical/oval shape, narrow size distribution <180 nm and positive zeta potentials (>20 mV) and good stability after one month of storage at 4 °C in formulation buffer or when incubated in culture medium and trypsin. In vitro studies of NPs cytotoxicity revealed that the elimination of formulation buffers led to an improvement in the rate of cell viability. The E7 antigen transcription was also increased for NPs obtained with high pDNA concentration (60 μg/mL). The analyzed CS-TPP-pDNA polyplexes can offer a promising vehicle for nucleic acid vaccines, not only in the prevention or treatment of viral infections, but also to fight emergent and future pathogens.

## 1. Introduction

Cervical cancer (CC) is considered one of the most common public health issues. Human papillomavirus (HPV) is regarded as the main carcinogenic pathogen, associated with CC. The current vaccines against HPV do not exert a therapeutic effect when the patient is already infected. Considering the relevance that nucleic acids vaccines have gained with the first approval for human use against COVID-19, DNA vaccines can provide a promising solution for the effective treatment of HPV-infected individuals, being fundamental to propose adequate methods to protect and deliver nucleic acids [1]. Nanocarriers have received huge attention in nanomedicine due to their ability of efficient drug delivery in several target cells. They exhibit some fascinating properties regarding limited steric obstruction, due to their nanometer size, and the ability to protect the therapeutic cargo, not only at the extracellular but also at the intracellular level [2]. Particles with uniform shape, ranging in size between one and one thousand nm and containing neutral or positive surface charge are considered adequate to carry drugs and efficiently deliver them into eukaryotic cells. The successful application of NPs is also determined by several desirable features, including their capacity to penetrate through several anatomical barriers, sustained and controlled release of their contents locally, their stability, and biocompatibility [3,4].

Nanoparticles are known to have a high surface/volume proportion and, the active ingredient or biopharmaceutical can be annexed to their surface or combined in their core or matrix [5]. Regarding the movement of drugs within the body, nanocarriers, in most cases offer many advantages, such as reduced clearance, extended circulation time, increased biological half-life, the possibility of surface modification for target delivery, and additionally increase the mean residence time (MRT) in the circulation system [5]. Several materials, such as liposomes, dendrimers, peptides, and polymers (synthetic or natural origin), have been explored in the formulation of nanocarriers to improve drug delivery and scale down toxicity [3,5,6,7,8,9].

Among the diversity of NP applications in biomedicine, the complexation of nucleic acids for gene therapy or vaccine delivery has been considered an emergent approach [10]. The two most explored non-viral vectors for genetic vaccine delivery are cationic phospholipids and cationic polymers. These cationic materials can be easily and efficiently combined with negatively charged DNA by electrostatic interactions, resulting in lipoplexes or polyplexes, respectively [11]. Polyplexes, when compared with lipid-based NPs, present some advantages such as lower cytotoxicity, easy manipulation, higher stability, and more controlled release of target DNA [11]. Among the polyplexes, chitosan (CS) appears as one of the most frequently mentioned polymers in the life sciences research studies, handling an extensive range of biopharmaceutical and biomedical approaches and applications [12].

Chitosan is a natural linear cationic polymer composed of D-glucosamine and N-acetyl-D-glucosamine linked by β (1→4) glycosidic bonds and is considered one of the best alternatives to constitute polymeric NPs [13]. This polysaccharide shows good biological features for therapeutic applications in comparison to other materials, such as good biocompatibility, biodegradability, adsorption, poor toxicity, and mucoadhesion. Several studies also describe CS carriers as suitable non-viral vectors for gene delivery, considering the biological properties mentioned above and its cationic character, which favors nucleic acids encapsulation and cellular internalization [14,15]. CS properties, such as molecular weight, deacetylation degree (DDA), CS derivatives, the ratio of positively-chargeable polymer amine (N = nitrogen) groups to negatively-charged nucleic acid phosphate (P) groups, N/P ratio, must be taken into consideration when choosing the best polymer for efficient gene delivering [16]. Although CS applications have been explored with less extension for DNA vaccines, its mucoadhesion properties transform this particular polymer into a promising material to develop DNA vaccine delivery nanosystems, with the intention of introducing needle-free administration via mucosal direct application, for instance. This vaccination modality can be useful for vaccine administration against HPV through vaginal mucosa, since it is the primary entry route of this virus. Furthermore, vaccines which are delivered directly to the mucosal site can provide a safer and efficient strategy to elicit both systemic and mucosal immunity, compared to parenteral administration [17,18]. This mucosal vaccination approach will also reduce pain and stresses associated with needle-based injection, eliminate biohazards of needle-disposal and avoid the need of trained medical personnel, all beneficial to overcome the antivaccination movement and increase vaccination rate [18].

To enhance the CS gelation and reticulation ability, as well as the NP’s properties in terms of small size and uniform morphology, the use of cross-linking agents (physical or chemical) have been extensively explored. Reversible physical cross-linking by electrostatic interactions is applied in NP formulations to avoid the toxicity of chemical agents and to improve cell viability and drug integrity. Polyanions have been investigated as physical cross-linking agents, among them tripolyphosphate (TPP), a non-toxic salt with multivalent anions, which is considered one of the best alternatives and is the most used crosslinker for cationic polysaccharides such as CS [19]. However, TPP is considered a flocculant that will simultaneously bind to two particles and cause aggregation by “bridging” the particles together, giving origin to the aggregation mechanism during the formation of CS/TPP NPs [20,21]. In this way, a strict control of polydispersity index (PDI), flocculation and aggregation of NPs, formulated by ionotropic gelation method by adding TPP, should be considered, monitored, and evaluated to ensure pDNA encapsulation efficiency, cell transfection, gene expression and low cytotoxicity of CS/TPP NPs.

The present work thoroughly explored the necessary conditions for the ionotropic CS gelation, as a potential natural polymer and for the suitable encapsulation and delivery of a plasmid DNA (pDNA) vaccine against CC induced by HPV persistent infection. The NPs were formulated by mixing a solution containing the pDNA and the anionic crosslinker TPP, with the CS cationic polymer, under constant magnetic stirring and preliminary conditions established by our research group [1] with slight modifications according to other works [22,23]. Thus, an exhaustive study was performed by manipulating fundamental parameters, especially the CS and TPP concentrations and volumes added during the ionotropic gelation method, to solve the issues associated with PDI, aggregation and cluster formation and to reach the best formulation of CS-TPP-pDNA polyplexes. The systems optimization and characterization was controlled to guarantee low PDI and avoid aggregation of NPs, ensuring suitable size distribution, pDNA encapsulation efficiency, particle shape and morphology, surface charge, stability and low cytotoxicity, to allow good cellular internalization, cell transfection and target gene transcription in human fibroblast (hFIB) and RAW 264.7 cell lines.

## 2. Material and Methods

### 2.1. Materials

Medical grade CS 95/1000 (with a molecular weight range between 200 and 500 kDa was) purchased from Heppe Medical (Berlin, Germany), TPP was obtained from Acros Organics (Geel, Belgium), 35% hydrochloric acid, sodium hydroxide palettes, and glacial acetic acid were all acquired from VWR (Prague, Czech Republic). The following solutions were freshly prepared by using deionized water from VWR (Prague, Czech Republic): 2 M HCl, 10 M NaOH, 1%, and 2% (*v*/*v*) acetic acid. GRS Taq DNA polymerase was purchased from Sigma Aldrich Chemicals (St. Louis, MO, USA). TripleXtractor used in RNA extraction was obtained from GRISP (Porto, Portugal). DMEM-F12 and DMEM-HG were purchased from GIBCO (Waltham, MA, USA). Sodium bicarbonate was obtained from MP Biomedicals (Santa Ana, CA, USA). Agarose, MgCl_2_ and GreenSafe were obtained from NZYtech (Lisbon, Portugal). All solutions were freshly prepared by using ultra-pure grade water, purified with a Milli-Q system from Millipore (Billerica, MA, USA).

### 2.2. Methods

#### 2.2.1. Plasmid Production, Extraction and Purification

The pMC.CMV-MCS-EF1-GFP-SV40PolyA pDNA vaccine, encoding human papillomavirus E7 protein next to CMV promoter, previously cloned by Serra and colleagues [24], was amplified in the *Escherichia coli* host. Briefly, the bacterial strains were inoculated in Luria-Bertani (LB)-agar plates containing 50 μg/mL of the kanamycin antibiotic and incubated overnight at 37 °C. After the growth step in the solid medium, some colonies were recovered and transferred to a 250 mL Erlenmeyer containing 62.5 mL of Terrific Broth (TB) liquid pre-fermentation medium (20 g/L of tryptone, 24 g/L of yeast extract, 4 mL/L of glycerol, 0.017 M of KH_2_PO_4_, 0.072 M of K_2_HPO_4_), supplemented with 50 μg/mL kanamycin. The cells were grown at 37 °C in a shaking orbital (Agitorb Aralab^®^ 200, Albarraque, Portugal) at 250 rpm. The optic density (OD) was measured until it reached 2.6 (corresponding to the exponential cell growth). At this moment, a specific volume of this medium containing cells was transferred to four 500 mL capacity Erlenmeyers with 125 mL of TB fermentation medium. These solutions were incubated at 37 °C, overnight (for approximately 18 h) at 250 rpm. After this period, bacterial growth was interrupted by cell centrifugation at 4500 rpm for 10 min at 4 °C and the pellets were stored at −20 °C until further use.

The pDNA extraction, recovery, and purification were obtained by using NZYTech Plasmid Maxi kit (Lisbon, Portugal), following the manufacturer’s protocol. The final pDNA sample was resuspended in 1 mL of 10 mM Tris-HCl buffer at pH 8.0. The pDNA concentration was measured at 260 nm using a NanoPhotometer™.

#### 2.2.2. Preparation of CS-TPP-pDNA NPs 

The method used to prepare the CS-TPP-pDNA polyplexes was the ionotropic gelation, using CS as polymer and TPP as crosslinker. Particles with and without pDNA were prepared following the process described in several works [1,22,23] with some adjustments. The CS stock solution was prepared in the concentration of 0.1% (*w*/*v*) in 1% (*v*/*v*) acetic acid, and the TPP stock solution was prepared in the concentration of 0.1% (*w*/*v*) in deionized water. The pH value of the CS and TPP stock solutions were adjusted to 5.2–5.5 and 2 by adding NaOH and HCl, respectively. Several dilutions of stock solutions were considered as well as different volumes of each solution to explore the best conditions for the nanoparticle formulation, keeping the pH values. Then, both CS and TPP solutions were filtered using a 0.20 μm polyethersulfone syringe filter from VWR (Radnor, PA, USA). For the experiments with pDNA encapsulation, the pDNA sample was dissolved into the TPP solution to reach the concentration of 20 μg/mL. A syringe pump (Harvard Apparatus 11 Plus, USA) was used to add TPP or TPP-pDNA solutions to the CS solution dropwise with a flow rate of 0.25 mL/min. To accomplish regular drop size, a needle size of 20 G was used in all experiments, leading to an addition rate of about 15–16 drops/min. During the addition, the CS solution was vigorously stirred (600 rpm) using a magnetic stirrer. After the complete addition of TPP or TPP-pDNA, the final solution was mixed for 30 min. All experiments were accomplished at room temperature, in triplicates.

#### 2.2.3. Indirect Encapsulation Efficiency (iEE)

To estimate the amount of pDNA encapsulated in the NPs, the NPs’ solution was centrifuged using Centrifuge Hettich Mikro 200 (Tuttlinggen, Germany) for 10 min at 10,000 rpm, and the DNA concentration in the supernatant was measured at 260 nm. The indirect encapsulation efficiency (iEE) was calculated following Equation (1), where c(total) is the initial pDNA concentration in the solution, and c(sup) is the measured pDNA concentration in the supernatant after the encapsulation procedure.
(1)$$\mathrm{iEE}\;\%\;=\frac{c\left(\mathrm{total}\right)-c\left(\sup\right)}{c\left(\mathrm{total}\right)}\times 100\%$$

#### 2.2.4. Particle Size, PDI and Charge Determination

The size PDI and charge of the prepared CS-TPP-DNA-polyplexes were also analyzed, using the Zetasizer Nano ZS analyzer (Malvern Instruments, Worcestershire, UK), equipped with a He-Ne laser by Dynamic Light Scattering (DLS). The particles were analyzed immediately following the preparation, after 72 h, and after one month to evaluate the stability of the particles in suspension. All DLS experiments were carried out at a temperature of 25 °C in triplicate and recorded into Zetasizer software v 7.03 (Malvern Instruments, Worcestershire, UK). This is the more suitable method to infer and compare the NPs sizes, since it is based on the frequency of movement and measures the hydrodynamic radii of the particles, which includes not only the particle itself but also the ionic and solvent layers associated with it in solution under the experimental defined conditions.

#### 2.2.5. Scanning Electron Microscopy (SEM) and Transmission Electron Microscopy (TEM)

The geometry and morphology of CS-TPP-pDNA NPs were evaluated by SEM. Freshly prepared systems were centrifuged (10,000 rpm, 10 min, 4 °C), the supernatant was discarded and the pellet was resuspended in phosphate-buffered saline (PBS) solution. Another centrifugation was performed in the same conditions, and the pellet was re-suspended in an aqueous solution containing 40 μL of 2 % tungsten. The samples were diluted at 1:10 in ultra-pure grade water. From the diluted solution, 10 μL was pipetted to a roundly shaped coverslip (10 mm) and left to dry overnight at room temperature. On the next day, samples were assembled on aluminum holders, attached with double-sided adhesive tape, and sputter-coated with gold using an Emitech K550 (London, UK) sputter coater. A scanning electron microscope, Hitachi S-2700 (Tokyo, Japan) with an accelerating voltage of 20 kV, was used and various magnifications were applied to evaluate the NPs morphology. TEM (Hitachi, Japan) was also performed after resuspension of nanoplexes in deionized water and sonication.

#### 2.2.6. Electrophoresis 

The agarose gel was prepared in the concentration of 1%, by diluting 0.4 g of agarose in 40 mL of TAE 1x buffer (40 mM Tris base, 20 mM acetic acid, 1 mM EDTA at pH 8.0), with the addition of 0.6 μL of green safe. An amount of 18 μL of each sample and 2 μL of loading buffer was added inside the well. The electrophoresis ran at 120 V for 40 min and the gel analyses were made through Uvitec Fire-Reader system (UVItec Limited, Cambridge, UK).

#### 2.2.7. Stability Assay

Two different stability assays were performed with the formulated nanoparticles. The first assay aimed to assess the NP’s colloidal stability and system properties over time, storing them in the formulation buffer at 4 °C after 72 h and one month. In the second assay, 250 μL of particles were centrifuged at 10,000 rpm for 10 min at 4 °C and the supernatant was analyzed by electrophoresis to evaluate the presence of free pDNA. Pellets were resuspended in 25 μL of DMEM-F12 medium supplemented with 10% of FBS and other with 25 μL of trypsin and incubated for 0, 2 and 6 h at 37 °C. These samples were evaluated by electrophoresis to determine the presence or degradation of DNA, which indicate the NPs decomplexation. Three controls were considered, namely the pDNA, pDNA incubated with DMEM-F12 medium, and pDNA incubated with trypsin. 

#### 2.2.8. Cell Culture–Seed and Transfection

Cell culture experiments were performed in two cell lines, hFIB (ATCC^®^ PCS-201-012™) and RAW 264.7 (murine macrophage cells, ATCC^®^ TIB-71™). Human fibroblast cells were grown in Dulbecco´s Modified Eagle´s Medium with Ham’s F-12 Nutrient Mixture (DMEM-F12) supplemented with 10% *v*/*v* heat-activated FBS, 2.438 g/L sodium bicarbonate, and 1% (*v*/*v*) of a mixture of antibiotics composed of penicillin (100 µg/mL) and streptomycin (100 µg/mL). RAW cells were grown Dulbecco´s Modified Eagle´s Medium with High Glucose (DMEM-HG) supplemented with 10% not inactivated fetal bovine serum (FBS), 1.5 g/L sodium bicarbonate and with 1% (*v*/*v*) the same mixture of antibiotics previously described. Cells were seeded in 25 cm² T-flasks at 37 °C in 5% CO_2_ humidified environment. For transfection studies, hFIB cells and RAW cells were seeded in 12-well plates at a density of 2.5 × 10^5^ cells/well and 2 × 10^5^ cells/well, respectively, in 1 mL complete medium/well until reaching 50–60% confluence. Before the transfection procedure, the medium was replaced by a medium without FBS and antibiotic supplementation (simple medium). The cells were transfected with particles dissolved in incomplete medium. After 6 h of transfection, the incomplete medium was replaced by complete medium.

#### 2.2.9. Cell Cytotoxicity

The cell viability was evaluated by the resazurin assay. The hFIB and RAW cells were seeded in a density of 1 × 10^4^ cells/well in a 96 well-plate. After 24 h of seeding the cells, the complete medium was changed to simple medium and the cells were transfected with the NPs. The simple medium was changed again to complete medium after 6 h of the transfection. After 24 h of the transfection, the medium was discarded and a mixture of 100 μL of complete medium + 20 μL of resazurin was added to each well. The plate was incubated in the dark for 4 h at 37 °C in 5% CO_2_ humidified environment. The volume of 100 μL of each well was transferred to an opaque plate and the fluorescence excitation at 544 nm and emission at 590 nm were read. The measurements were performed using a plate reader spectrofluorometer (Spectramax Gemini XS, Molecular Devices, San Francisco, CA, USA). The same procedure was repeated after 48 h of transfection.

#### 2.2.10. Detection and Assessment of E7 mRNA Transcripts

##### Total RNA Extraction

The hFIB and RAW cells were seeded in a density of 2.5 × 10^5^, per well in 12-well plates. To extract the total RNA, the medium inside the wells was removed and the well was washed twice with PBS. The cells were lysed through the addition of 200 μL of TRIZOL (Thermo Scientific, Lisbon, Portugal) to each well. Homogenization was performed with the pipette by doing “up and down” with the liquid inside the well until the appearance of a viscous solution. Samples were incubated for 5 min at room temperature. Chloroform was added (50 μL) to each sample and vigorously mixed by inversion to separate different biomolecules. Samples were incubated for 10 min at room temperature and then were centrifuged at 4 °C for 15 min at 12,000 rpm. After the centrifugation, two different layers were obtained. The top layer containing the aqueous phase was gently recovered to avoid the destabilization and contamination of the RNA phase. The RNA precipitation was performed by adding 125 μL of cold isopropanol to the lower layer, carefully mixing by inversion, incubating on ice for 10 min, and centrifuging at 4 °C for 15 min at 12,000 rpm. The supernatant was removed, and the pellet was resuspended in 125 μL 75% ethanol (prepared in diethylpyrocarbonate (DEPC) water) to eliminate the organic compounds, and samples were centrifuged at 4 °C for 5 min at 12,000 rpm. The supernatant was discarded, and the pellet was dried for 5 min and rehydrated with 20 μL of DEPC water. The RNA was quantified using the the NanoPhotometer™.

##### cDNA Synthesis

The cDNA synthesis was performed with Xpert cDNA Synthesis kit (GRiSP Research Solutions, Porto). A mixture of 1 μg of the RNA sample obtained above, 1 μL of deoxynucleotides (dNTPs), 1 μL of random primers, and addition of RNase free water until reaching the final volume of 14.5 μL was prepared in an RNase free tube. The tubes were incubated at 65 °C for 5 min and then into ice for 2 min. Thereafter, 4 μL of Xpert buffer, 0.5 μL of RNase Inhibitor and 1 μL of Xpert RTase were added to the prepared mix and gently homogenized. The samples were placed into the T100™ Thermal Cycler (Bio-Rad Laboratories, Inc, Hercules, CA, USA) at 25 °C for 10 min, 50 °C for 50 min, and finally at 85 °C for 5 min. After the cDNA synthesis, samples were used to perform the RT-PCR assay or stored at −20 °C until further use.

##### Reverse Transcription Polymerase Chain Reaction (RT-PCR)

Qualitative evaluation of E7 mRNA transcripts was conducted by RT-PCR. For the PCR experiment, a mixture of 0.7 μL of magnesium chloride (MgCl_2_), 0.25 μL of deoxynucleotides (dNTPs), 0.40 μL of forward primer (5′−AAT CTA GAA TGC CTG ATA CAC CTA C −3′) and reverse primer (5′ −ATG GAT CCT TAT GGT TTC TGA GAA CAG A −3′), 1.25 μL of buffer, 1 μL of the cDNA sample synthesized as mentioned above, 0.25 μL of GRS Taq DNA polymerase and 8.25 μL of RNase free water. Samples were then placed into the T100™ Thermal Cycler, with the subsequent settings: 95 °C for 5 min, 26 cycles of 30 s at 95 °C, 30 s at 60 °C and 1 min at 72 °C, finally 10 min at 72 °C and to finish the amplification the samples were put into 4 °C. Final samples were analyzed by electrophoresis and visualized in the Uvitec Fire-Reader system.

##### Reverse Transcription Quantitative Real-Time PCR (RT-qPCR)

The level of E7 mRNA transcripts was also quantitatively analyzed by RT-qPCR. The mix for a reaction with primers designed for the E7 gene transcript was prepared with 10 μL of SYBR ™ Green Master Mix, 0.64 μL forward primer, 0.64 μL reverse primer, 7.72 μL of sterile H_2_O and 1 μL of cDNA, resulting in a volume of 20 μL per reaction. The mix for a reaction with the primer pair of the GAPDH housekeeping gene transcript (FW: 5’− ATG GGG AAG GTG AAG GTC G −3 ‘; RV: 5’− GGG GTC ATT GAT GGC AAC AAT A -3’) was prepared with 10 μL of SYBR ™ Green Master Mix, 1.2 μL FW primer, 1.2 μL RV primer, 6.6 μL of sterile H_2_O and 1 μL of cDNA, reaching a final volume of 20 μL. The reaction mixtures were placed in a Real-Time CFX ConnectTM system (BioRad, Hercules, CA, USA), programmed with the following sequence of incubations: 10 min at 95 °C and 40 cycles of 15 s at 95 °C, 30 s at 60 °C.

#### 2.2.11. Statistical Analysis

Each experience was performed at least three times. Data are expressed as a mean ± standard error (S.D.). The statistical analysis performed was a one-way and two-way analysis of variance (ANOVA), followed by Tukey’s test. Data analysis was performed in GraphPad Prisma 6 software. A *p*-value below 0.05 was considered statistically significant. Additionally: * *p* < 0.05; ** *p* < 0.01; *** *p* < 0.001; **** *p* < 0.0001.

## 3. Results and Discussion

Based on a preliminary study regarding the preparation and formulation of the CS-TPP and CS-TPP-DNA polyplexes performed by our research group, some prior experiments were conducted to achieve the best formulation for use [1]. Obtained results showed very small particles, below 40 nm, by changing the DNA concentration but with PDI values above 0.45. This study exhibited very good results regarding the particles size, although the PDI of formulations was not ideal in all cases [1]. Low PDI suggests more homogeneous NPs and high PDI indicates non-uniformity formulations, flocculation and aggregate formation, resulting in broad particle size distribution [25]. In fact, aggregation of chitosan nanoparticles is a well-known drawback of these systems, which in most works performed some years ago, were neither controlled nor the PDI assessed. The clusters distribution can present a problem regarding the cell internalization, even with small particles. To overcome these issues associated with PDI, some previous works used an ultra-sonication step during the NPs production or the addition of a surfactant such as Tween 80 to avoid NPs aggregation [26,27]. However, the encapsulation efficiency of their systems was very low (below 40%), which indicates that the ultra-sonication process disrupts the NPs, releasing part of the encapsulated content, and diminishing the encapsulation efficiency rates or the presence of detergents increases the NPs toxicity. Thus, expecting to obtain reasonable results of particle shape and morphology, size distribution, surface charge, and pDNA encapsulation efficiency to ensure cell transfection, gene expression and low cytotoxicity, the issues associated with PDI, aggregation and cluster formation needed to be improved. Thus, in the present work an exhaustive study was performed regarding optimization of some fundamental parameters, especially the CS and TPP concentrations and volumes added during the ionotropic gelation method, to reach the ideal formulation system.

### 3.1. Influence of Changing TPP Volume on NPs Size, Charge, and PDI 

Al-Nemrawi and colleagues used the ionotropic gelation method to obtain particles ranging from 145 to 663 nm [28]. They proved that reducing the amount of added TPP decreases the particle size, while a a positive global charge is maintained. In addition, they showed that when the TPP volume increased, the PDI also increased. Increasing the TPP addition rate from 0.25 to 2.5 mL/min led to an increase of PDI value from 0.08 to 0.47, respectively. In a study with medium molecular weight (MMW) and high molecular weight (HMW) CS polymers, Zaki and colleagues observed the CS concentration also can directly affect the NPs size, i.e., the increase of CS and TPP concentration resulted in an increase of NPs size [29]. Following their work, the changing these two parameters, CS and TPP volumes and concentrations was further explored in the present work. First, the TPP volume (0.5 to 2 mL) was changed in three different formulations presented in Table 1. Physico-chemical properties of these formulations, such as size, PDI and charge, determined by DLS and Zetasizer, were recorded and are summarized in Figure 1.

As shown in Figure 1A, changing the TPP volume from 0.5 to 1 mL did not reveal significant changes to particle size, maintaining the values at around 180 nm in samples F1 and F2. In the formulation F3, an increase of TPP volume to 2 mL, resulted in particle sizes of almost 1000 nm. The formation of aggregates/flocculation probably led to this increase, and clusters were also observed inside the solution. The same effect was obtained by Tzeyung and co-authors in their experiments, keeping the drug concentration of 0.05%, CS and TPP concentrations of 0.05%, and varying the TPP volume (5, 6, and 7 mL). An increase of 10 nm was observed from 5 to 6 mL of added TPP; however, upon increasing the volume to 7 mL a significant difference of 98 nm was noticed [30].

Concerning the global charge analyzed by zeta potential, formulation F1 showed the best result (around 24 mV). Given that TPP is negatively charged, when its volume is increased (formulations F2 and F3), the NPs global charge decreases to 17 and 8 mV, respectively (as depicted in Figure 1B). The same pattern was observed by Rodolfo and co-authors, since with the increase in TPP concentration and volume, the zeta potential decreased in all CS polymers used. Their work showed that TPP could be responsible for decreasing the zeta potential values due to the increment of the global negative charge of the system [31]. The positive surface charge of carriers is a very important requirement to improve the cell uptake, and consequently reach an efficient gene delivery system. Given the cell membranes are negatively charged, the opposite charges of NPs benefit the electrostatic attraction between them and improve the internalization [11]. 

Considering the size of samples F1 and F2 (Figure 1A), small particles are observed (<200 nm). However, regarding the PDI depicted in Figure 1C, polydisperse particles are formed, showing high/non-satisfactory PDI values (>0.3). A significant difference of the PDI value can be detected by decreasing the TPP volume, reaching values below 0.35 in the formulation F1 when 0.5 mL of TPP was used. Koping-Hoggard and colleagues also observed a decrease in the PDI to 0.25 with the reduction of TPP volume [32].

### 3.2. Influence of Changing CS Volume and TPP Concentration on NPs Size, Charge, and PDI

According to previous results, the PDI was improved by decreasing the TPP volume to 0.5 mL, suggesting this parameter should be sustained. Thus, new formulations are described in Table 2 by maintaining the TPP volume and manipulating the CS volume and TPP concentration. Moreover, physico-chemical properties of these formulations, such as size, PDI, and charge, determined by DLS and Zetasizer, were recorded and are presented in Figure 2.

Concerning the NPs size, obtained results showed that the CS volume variation (3 and 4 mL) did not significantly affect this parameter. However, when decreasing the concentration of TPP from 0.1 to 0.05%, smaller particles are obtained (Figure 2A). This behavior was also observed by Bangun and colleagues, since they verified that the increase of the TPP concentration resulted in larger particle sizes and led to precipitation or sedimentation of NPs in solution. In their work, they observed the appearance of the precipitated samples by the Fourier-transform infrared spectroscopy (FTIR) technique [33]. Huang and Lapitsky also showed that aggregation kinetics were slow at low TPP concentrations and low pH values, avoiding the NPs precipitation and formation of clusters/flocculation [20]. 

As shown in Figure 2B, the NPs charge does not significantly change by increasing or decreasing the CS volume and TPP concentration. Moreover, reducing the CS volume from 4 to 3 mL, a decrease in PDI value is observed from 0.34 to 0.25, as shown in Figure 2C. When the TTP concentration is reduced, the same effect is noticed, a decrease from 0.39 to 0.32 in PDI values, which means more monodisperse nanoparticles are obtained. The negative impact of higher TPP concentration on PDI was also observed by Rodolfo and colleagues, which concluded that this parameter should be controlled because it is directly related to aggregation phenomena [31].

### 3.3. Last Adjustments to Reach Good Formulations with and without pDNA

Nanoparticles with and without pDNA (20 μg/mL) were formulated based on the results described in Section 3.2 and including the following adjustments: using low CS concentration (0.02%), different added volumes (2 and 3 mL), and low volume of TPP (0.25 mL) at 0.05 %. The formulations and obtained physico-chemical properties are summarized in Table 3.

The chosen conditions revealed very satisfactory results, obtaining sizes between 120 and 180 nm, charges around 20 mV and all formulations were monodisperse with a PDI values below 0.25. An improvement of iEE was observed when the CS volume was increased from 2 to 3 mL (iEE around 78%), suggesting that 3 mL of CS is more suitable to ensure the pDNA encapsulation. 

The NPs size of all systems was lower than 180 nm. These results can be explained by the volume of TPP solution used (0.25 mL). As Rampino and colleagues had also verified in their study, NPs aggregation occurred while using a high volume of TPP with a low volume of CS. They confirmed that increasing the amount of TPP into the CS solution led to flocculation or particle packing [34]. The same behavior was noticed by Masarudin and colleagues, where they showed that above 0.2 mL of TPP addition, the particle size and PDI significantly increased. They tested three different formulations, and concluded that above 0.20 mL of TPP addition, PDI values substantially increased, reaching values above 0.60, evidencing very polydisperse particles, presence of aggregated systems, and consequently high NPs size [21].

The decrease of NPs size in Table 3 can also be explained by the reduction in CS concentration from 0.035% to 0.02%. The same behavior was observed by Zaki and co-authors, since they evaluated the effects of different CS concentrations on the particle size prepared from MMW and HMW CS [29]. For MMW, the particle size was increased with the increase of the CS concentration from 0.2% to 0.6% *w*/*v*. The particle size of HMW NPs was also significantly increased from 987 nm to 1651 nm when CS concentration was increased from 0.2% to 0.3% *w*/*v* [29]. Tzeyung and co-authors also observed an increase in particle size when the CS concentration was increased (from 0.05 to 0.15% *w*/*v*). They correlated this tendency with the high amount of CS chains per volume, thus forming large particles when the TPP cross-linking agent was added. This difference can also cause a decrease in the TPP cross-linking density by the CS, resulting in particle aggregation and formation of larger particles [30]. 

Regarding the zeta potential, no significant difference was observed after the addition of pDNA in both formulations, with charge values of approximately 20 mV. Similar results concerning zeta potential values were found by Carrillo and co-authors, observing only a slight decrease between the particles with and without pDNA, although all formulations showed positive zeta potential values [35]. In addition, the NPs charge did not significantly change by decreasing the CS concentration from 0.035% to 0.02%. The same pattern was observed by Zaki and co-workers, where they analyzed the zeta potential of NPs formulated with MMW and HMW was not significantly affected by changes in the CS concentration [29].

### 3.4. Reproducibility and Storing Stability Assessment

Three independent experiments were prepared to formulate NPs with and without pDNA (20 μg/mL), applying 3 mL of 0.02% CS and 0.25 mL of 0.05% TPP, to confirm the reproducibility of ionotropic gelation technique and the stability of NPs during their storage at 4 °C, in the formulation buffer over the time (0, 72 h and 1 month). The respective results are presented in Table 4.

All formulations were prepared with fresh solutions of CS, TPP, and pDNA when applicable (formulations 4 to 6 in Table 4). The achieved NPs showed that the ionotropic gelation technique is reproducible since particles with and without plasmid DNA formulated in three replicates share very similar size, PDI, and zeta potential. As expected, a small increase in the NPs size is noticed after adding the pDNA. However, the size was still below 180 nm, and the PDI was around 0.20, showing monodisperse particle population. Da Silva and colleagues presented very similar results, using a low amount of TPP inside the CS solution, in acidic pH (5.8), in which they displayed NPs sizes in the range of 200 to 300 nm, with zeta potential values around 20 and 30 mV [36].

According to Sreekumar and co-authors, reproducibility of the system still presents a problem that compromises the promising market applications of CS nanoparticles formed by the ionotropic gelation method [37]. The initial CS concentration and the solvent atmosphere of the CS solution were considered the two main factors to create a reproducible system. They were also able to control the NPs size by comprehending and manipulating these two factors [38]. Elgadir and colleagues also found in other studies the feasibility of reproducible and stable NP systems that can encapsulate and deliver drugs. They mentioned nanomedicine as a novelty regarding cancer treatment, diagnosis, and detection, and affirmed that polymeric nanoparticles are promising drug carriers. Their study showed in vitro and in vivo experiments that support and sustain their opinion [38].

No significant differences were noticed in zeta potential values of NPs formulated with and without pDNA and stored throughout the time, present in Table 4, represented by the values around 20 mV, which suggests the system’s stability. High zeta potential values indicate that they will not attract each other due to repulsion of the positive charges created between them. In this way, aggregation or formation of clusters is avoided. Nevertheless, if the zeta potential value of particles is not high enough to avoid the attraction forces between them, precipitates can be formed leading to unstable systems, not only at the moment of the complexation but also after some days or weeks of storage [39].

The NPs stability was checked by measuring system properties after 72 h and one month. Even after one month of storage at 4 °C in the formulation buffer, the solution remained clear and no aggregation or precipitation was noticed. Size, PDI, and zeta potential shown in Table 4 confirm that there are no significant variations throughout the time of the NPs storage. After the analysis of all parameters, pDNA (20 μg/mL), 3 mL of 0.02% CS and 0.25 mL of 0.05% TPP conditions were chosen as the best to proceed for in vitro transfection studies with this pDNA delivery system. 

### 3.5. Polyplexes Morphology

The NPs shape can directly influence their internalization by the cells. The surface morphology of CS-TPP-pDNA polyplexes were analyzed by scanning electron microscopy (SEM) and transmission electron microscopy (TEM). Figure 3 shows images of CS-TPP-pDNA nanocarriers developed in this work. 

By SEM and TEM analysis, it can be observed that the formulated NPs presented in Figure 3 show a spherical or oval shape and uniform morphology, with sizes less than 200 nm, and that are suitable for cellular uptake and internalization. Several studies described in the literature, highlight the spherical and oval NPs shape and its benefits in comparison to rod shapes, especially showing high cellular uptake/transfection efficiency. Multiple authors have shown that NPs with spherical morphology exhibited greater cellular internalization in comparison to rod-shaped NPs [40,41]. 

### 3.6. Stability Tests

The stability tests were performed with CS-TPP-pDNA NPs formulated under the best conditions. The assay was performed at three different incubation times (0, 2 and 6 h) to evaluate the behavior of naked pDNA and NPs in contact with DMEM-F12 medium supplemented with 10% of serum (FBS) and trypsin. These incubation conditions simulated the in vitro cell transfection and in vivo extracellular conditions to ensure the success of cellular transfection of these systems [31]. The agarose electrophoresis was used to analyze the supernatant of each sample, after the formulation and after the incubation time (Figure 4), to verify the presence of free plasmid, which indicates low EE during the formulation or the NPs decomplexation after the incubation.

In the results present in Figure 4A1,A2, no free DNA is observed in the agarose gel for all supernatants of the performed experiments, evidencing the amount of the added plasmid which should be entrapped into the CS-TPP-pDNA polyplexes. Similar results were also reported by Carrillo and co-workers, once no free DNA was observed in the agarose gel, even for different CS concentrations and for different plasmid volumes [36].

After the incubation with culture medium, continuous degradation of naked pDNA is observed along the time in lanes 2, 3 and 4 of Figure 4B1. However, no free DNA is observed in the lanes where pDNA encapsulated in nanocarriers is tested. These results suggest that the CS-TPP-pDNA NPs remained stable in these incubation conditions and effectively protected the plasmid from degradation for at least 6 h. Similarly, Figure 4B2 evidences that no disruption happens in NPs during 6 h of incubation with trypsin, due to the non-appearance of pDNA bands in electrophoresis image. 

### 3.7. Cell Viability Evaluation

Cellular cytotoxicity is an important issue to be analyzed when choosing the best delivery system to ensure its suitability for cellular transfection in vitro and in vivo assays. The resazurin cell viability assay, also called Alamar blue, is a fluorescent experiment to identify the cell metabolic activity [42]. In this context, to determine toxicity levels of formulated NPs, in function of the level of formulation buffer impurities removed by centrifugation and using 20 or 60 µg/mL of pDNA, the resazurin assay was performed on hFib and RAW cells for 24 and 48 h (results present in Appendix A). The results in hFIB cells revealed that the elimination of formulation buffer components is fundamental to avoid their influence in cell viability. The results in RAW cells confirmed non-significance between control and all prepared systems, supporting that no cytotoxicity was induced. Valente and co-authors compared the cytotoxicity between two systems (polyethylenimine (PEI) and CS) in HeLa cancer cells and hFIB, and proved that CH-based polyplexes loaded with different pDNAs did not lead to cytotoxicity and also recommended CS-based systems as safer for pDNA delivery than PEI-based systems [11]. Da Silva and colleagues also prepared CS NPs using the ionotropic gelation technique to encapsulate rosmarinic acid. The particles showed low cytotoxicity against the retinal pigments and the corneal cells. They concluded that CS NPs are a promising approach for drug release in ocular applications [36].

### 3.8. E7 Gene Expression

After transfection of hFIB and RAW cells with the developed gene delivery systems, E7 mRNA expression was evaluated. Firstly, total RNA was extracted, then the RNA was reverse transcribed into cDNA, and RT-PCR experiment was performed to amplify the E7 gene using specific primers. All products of the obtained PCR were visualized by electrophoresis in 1% agarose gel. For both experiments, three different pDNA concentrations were used 20, 40 and 60 μg/mL (Figure 5). 

E7 mRNA transcripts were amplified in all cells transfected with CS-TPP-pDNA systems. No band was observed in the second lane (Control), indicating no contamination occurred during the assays, and in the third lane suggesting non-transfected cells did not present the E7 gene expression. The level of mRNA transcripts seems to increase in both cell lines with an increase of the pDNA concentration. The RT-PCR technique demonstrated that the produced nanoparticles had the desired and intended effect. The formulated systems show excellent protection of the pDNA through the transfection process and arrival to the nucleus. 

An RT-qPCR experiment was performed to quantify E7 mRNA expression levels. Considering this assay is more accurate and specific, it was applied to compare and evaluate the performance of two samples formulated with 60 μg/mL of pDNA but resultant from two different centrifugation speeds (6000 and 10,000 rpm) (Figure 6).

Results present in Figure 6 evidence an increase in the expression of E7 transcripts in relation to non-transfected cells (control) for both systems. CS-TPP-pDNA polyplexes centrifuged at 10,000 rpm show higher levels of E7 transcripts than polyplexes centrifuged at 6000 rpm, proving that the elimination of formulation buffer impurities can play a crucial role in the cell viability and, consequently, the degree of cellular internalization of polyplexes. The results suggest that these conditions are optimal for an efficient cellular internalization/uptake and consequent gene expression and is expected to induce higher protein expression. In addition, we intend to present a future study soon, consisting of the modification of these NPs with specific ligands, such as mannose, to provide a targeted delivery to antigen presenting cells in order to increase the specificity of the vaccine and consequently the intended immune responses by in vivo studies after mucosal administration.

## 4. Conclusions

The ionotropic gelation technique was employed in this work to produce polyplexes capable of encapsulating, protecting, delivering pDNA vaccines and maintaining the structural stability of the developed system. The obtained results revealed that many parameters have to be taken into account when developing these gene delivery vehicles, PDI in particular, to ensure the NPs homogeneity and to avoid the flocculation and aggregation phenomenon, which constitutes a problem regarding cell internalization, even with the formulation of small particles. Among the different conditions and parameters, the volume and concentration of CS and TPP revealed to have the most important role in the design of the NPs size, PDI, encapsulation efficiency, surface charge and morphology. The best conditions for the CS-TPP-pDNA formulation were achieved using 3 mL of 0.02% CS and 0.25 mL of 0.05% TPP. The resultant nanocarriers present a high pDNA encapsulation efficiency rate, nanometric sizes (<180 nm) and very homogeneous samples (PDI < 0.2), spherical/oval shape, and positive surface charge (>20 mV), suitable characteristics for pDNA vaccine delivery protocols. These NPs remained stable, even after one month of storage at 4 °C in the formulation buffer, and were not destabilized when incubated in culture medium and trypsin. The elimination of formulation constituents is fundamental to avoid cell toxicity, which also reflects in a high transfection efficiency and consequently target gene expression. These systems, based on CS and TPP, offer remarkable potential for the conceptual design and formation of pDNA vaccine delivery vectors towards the most demanding biomedical applications.

## Figures and Tables

**Figure 1 polymers-14-01443-f001:**
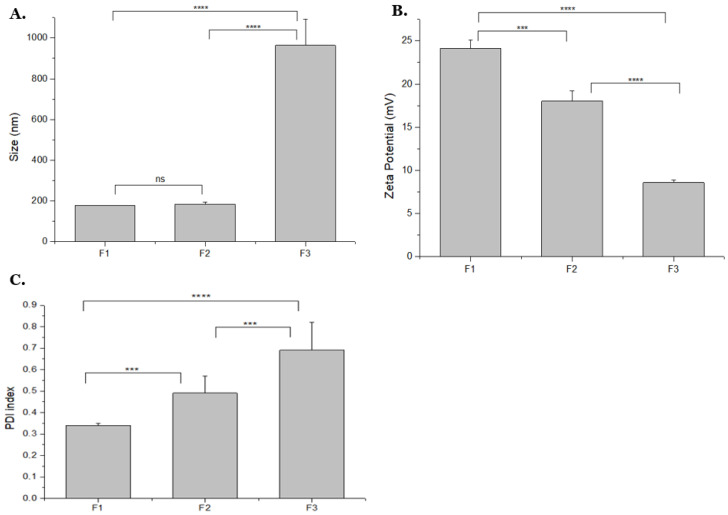
NPs (**A**) mean size (nm), (**B**) Average zeta potential, (**C**) PDI, obtained when different TPP volumes were added to formulations (F1 = 0.5; F2 = 1 and F3 = 2 mL). The values were calculated with the data obtained from three independent measurements (mean ± S.D., *n* = 3); *** *p* < 0.001; **** *p* < 0.0001, ns—not significant.

**Figure 2 polymers-14-01443-f002:**
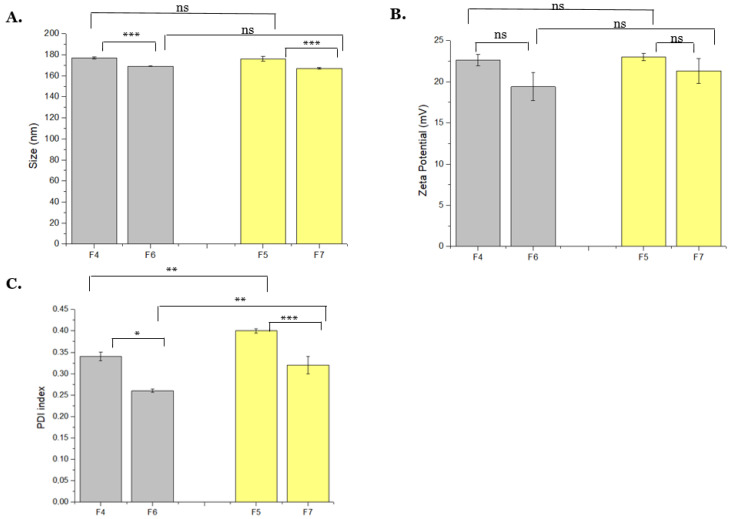
NPs (**A**) mean size (nm), (**B**) average zeta potential, (**C**) PDI, changing CS volumes and TPP concentrations (F4 to F7). The values were calculated with the data obtained from three independent measurements (mean ± S.D., *n* = 3) * *p* < 0.05; ** *p* < 0.01; *** *p* < 0.001; ns—not significant. Gray–CS volume of 3 mL, Yellow–CS volume of 4 mL.

**Figure 3 polymers-14-01443-f003:**
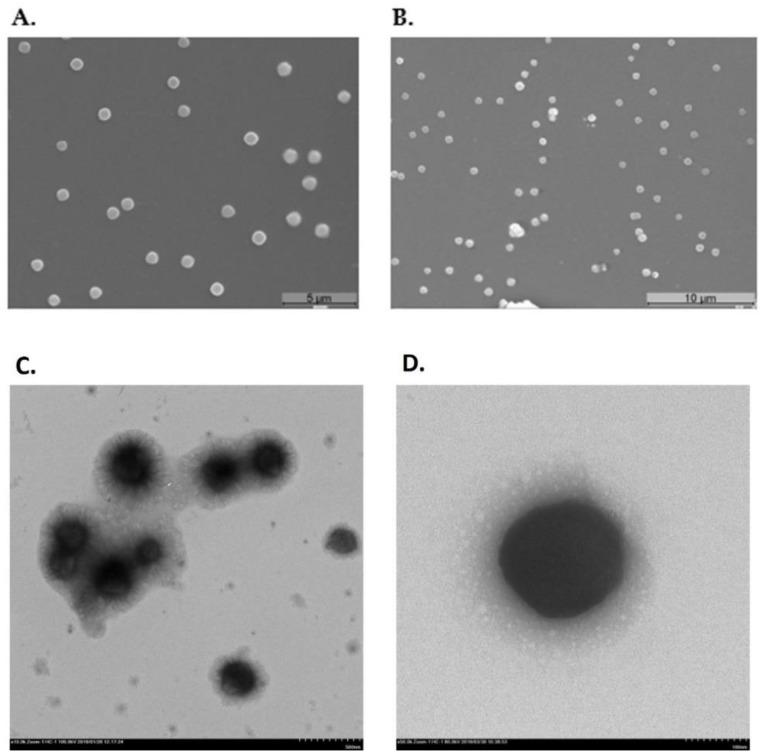
(**A**,**B**) Scanning electron micrographs of CS-TPP-pDNA NPs at magnification of 5000× and 3500×, respectively. (**C**,**D**) Transmission electron micrographs of CS-TPP-pDNA NPs.

**Figure 4 polymers-14-01443-f004:**
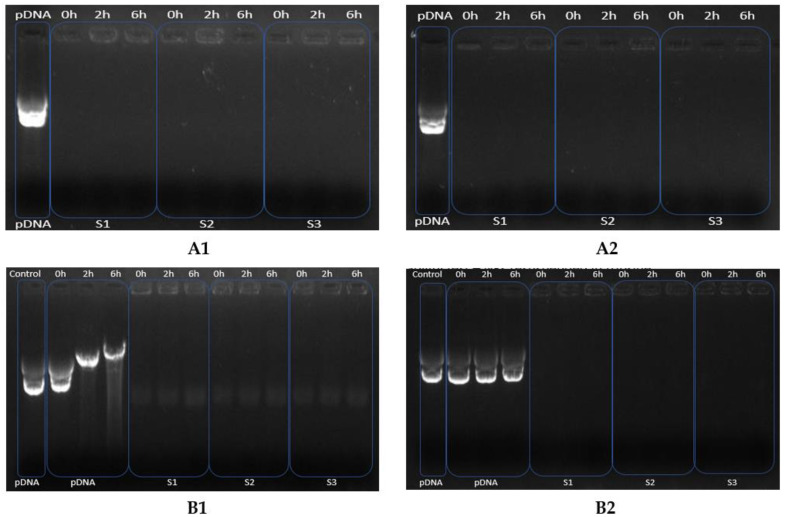
Electrophoretic analysis of NPs supernatants after the formulation step and before the NPs incubation with medium (**A1**) and trypsin (**A2**). Lane 1: pDNA control, lane 2–10: non-encapsulated pDNA samples 1, 2 and 3, incubation of 0, 2 and 6 h. Electrophoretic analysis of the system’s protection of pDNA after its incubation with DMEM-F12 medium supplemented with 10 % of serum FBS (**B1**) and after its incubation with trypsin in (**B2**). Lane 1: pDNA control; lane 2–4: pDNA with medium, at 0, 2 and 6 h. Lane 5–13: CS-TPP-pDNA sample 1, 2 and 3 with medium at 0, 2 and 6 h; Figure B2: same order, with trypsin.

**Figure 5 polymers-14-01443-f005:**
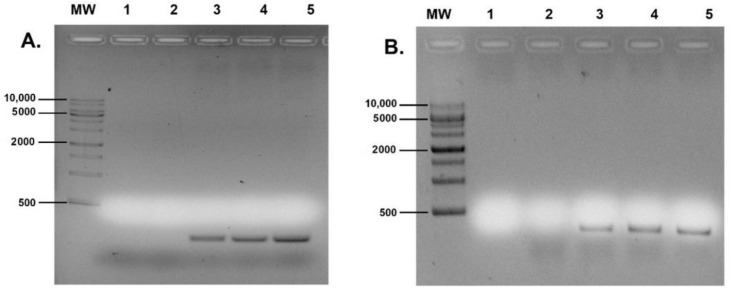
Analysis of RT-PCR products by agarose gel electrophoresis. Evaluation of E7 transcripts in hFIB cells (**A**) and RAW cells (**B**). Lane MW: DNA molecular weight marker (range from 500 to 10,000 bp); lane 1: control without cDNA sample; lane 2: non-transfected cells; lane 3: cells transfected by CS-TPP-pDNA [20 μg/mL]; lane 4: cells transfected by CS-TPP-pDNA [40 μg/mL]; lane 5: cells transfected by CS-TPP-pDNA [60 μg/mL]. Lanes 3, 4 and 5 present a band of 297 bp corresponding to the E7 transcripts.

**Figure 6 polymers-14-01443-f006:**
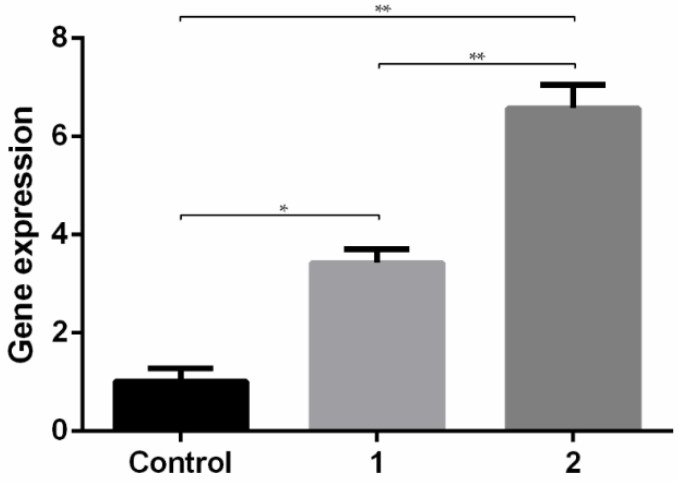
RT-qPCR of E7 expression levels in RAW cells. Control–non-transfected cells; 1–CS-TPP-pDNA, centrifugation speed 6000 rpm, pDNA concentration of 60 μg/mL; 2–CS-TPP-pDNA, centrifugation speed 10,000 rpm, pDNA concentration of 60 μg/mL. Data obtained from three independent measurements (mean ± S.D., *n* = 3). * *p* < 0.05; ** *p* < 0.01.

**Table 1 polymers-14-01443-t001:** Formulations of CS-TPP NPs by changing the added TPP volume.

Formulation Codes	CS Concentration (%)	CS Volume (mL)	TPP Concentration (%)	TPP Volume (mL)
F1	0.035	5	0.1	0.5
F2	0.035	5	0.1	1
F3	0.035	5	0.1	2

**Table 2 polymers-14-01443-t002:** Formulations of CS-TPP NPs by changing CS volume and TPP concentration.

Formulation Codes	CS Concentration (%)	CS Volume (mL)	TPP Concentration (%)	TPP Volume (mL)
F4	0.035	3	0.1	0.5
F5	0.035	4	0.1	0.5
F6	0.035	3	0.05	0.5
F7	0.035	4	0.05	0.5

**Table 3 polymers-14-01443-t003:** Formulations and respective characterization results of CS-TPP and CS-TPP-pDNA obtained with 0.25 mL of TPP (0.05%) and varying the CS (0.02%) volume.

Formulation Codes	CS Concentration (%)	CS Volume (mL)	Z–Average Size (nm)	PDI	Zeta Potential (mV)	Encapsulation Efficiency (%)
F8	0.02	3	141 ± 3.96	0.24 ± 0.030	20.3 ± 0.92	Without pDNA
F9	0.02	2	121 ± 0.58	0.23 ± 0.011	20.5 ± 1.04	Without pDNA
F10	0.02	3	167 ± 1.35	0.23 ± 0.008	20.2 ± 0.76	78 ± 11.7
F11	0.02	2	179 ± 0.60	0.17 ± 0.004	20.0 ± 0.65	55 ± 9.3

**Table 4 polymers-14-01443-t004:** Properties of NPs prepared in three independent samples with the best formulation conditions achieved previously, to check the reproducibility of the formulation technique and the NPs stability over the time. Formulations 1 to 3 without pDNA, 4 to 6 with pDNA.

	Formulation Numbers	Z–Average Size (nm)	PDI	Zeta Potential (mV)
**Particles Freshly Prepared**	1	140 ± 1.75	0.23 ± 0.015	20.2 ± 1.32
	2	146 ± 1.98	0.24 ± 0.019	20.3 ± 0.43
	3	158 ± 3.02	0.22 ± 0.009	20.4 ± 0.40
	4	172 ± 1.74	0.20 ± 0.008	21.7 ± 1.00
	5	178 ± 1.19	0.17 ± 0.007	19.9 ± 0.38
	6	175 ± 2.30	0.19 ± 0.016	19.6 ± 0.47
**After 72 h**	1	157 ± 4.59	0.25 ± 0.017	23.4 ± 0.38
	2	153 ± 1.24	0.26 ± 0.021	18.9 ± 3.20
	3	168 ± 3.79	0.28 ± 0.011	21.2 ± 1.12
	4	176 ± 0.94	0.24 ± 0.006	19.5 ± 0.78
	5	185 ± 0.87	0.22 ± 0.012	20.3 ± 0.80
	6	182 ± 3.14	0.20 ± 0.006	18.1 ± 1.08
**After 1 Month**	1	164 ± 6.32	0.24 ± 0.013	21.9 ± 0.24
	2	155 ± 1.24	0.25 ± 0.009	19.7 ± 0.63
	3	179 ± 3.79	0.26 ± 0.022	21.0 ± 1.07
	4	177 ± 0.94	0.27 ± 0.008	22.3 ± 0.45
	5	197 ± 0.87	0.21 ± 0.021	18.7 ± 0.59
	6	188 ± 3.14	0.22 ± 0.003	20.1 ± 0.22

## Data Availability

The data presented in this study are available on request from the corresponding author.

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
