# Peer review of "Modulation of Chitosan-TPP Nanoparticle Properties for Plasmid DNA Vaccines Delivery"

_polymers, 2022, doi:10.3390/polym14071443_

Round 1

Reviewer 1 Report

Nunes et al. reported the preparation of CS-TPP-DNA nanoparticles. The particle size, dispersity, and surface charge of the nanoparticles were optimized by varying the CS and TPP volume and concentration. After confirming the stability of the nanoparticles, they were evaluated by in vitro studies. The nanoparticles collected at high centrifugation speed showed reasonably low cytotoxicity and good target gene expression. The study is systematically performed and the observations support the conclusions. I recommend publication of the manuscript after some revision.

1. Polyplex morphology was examined using SEM in Figure 3. The more appropriate analysis should be done with TEM. Please consider adding some TEM images to show the nanoparticle size and dispersity.

2. Figure 6 shows rather poor gel electrophoresis images. Based on the data shown, it is not possible to confirm the band size. First, the ladder needs to be marked to show size information. Second, please consider rerun the gel so that the PCR bands are not located at the bottom of the gel. Especially in panel B, the bands are also cut off.

Author Response

Nunes et al. reported the preparation of CS-TPP-DNA nanoparticles. The particle size, dispersity, and surface charge of the nanoparticles were optimized by varying the CS and TPP volume and concentration. After confirming the stability of the nanoparticles, they were evaluated by in vitro studies. The nanoparticles collected at high centrifugation speed showed reasonably low cytotoxicity and good target gene expression. The study is systematically performed and the observations support the conclusions. I recommend publication of the manuscript after some revision.

  1. Polyplex morphology was examined using SEM in Figure 3. The more appropriate analysis should be done with TEM. Please consider adding some TEM images to show the nanoparticle size and dispersity.

Answer 1: We thank the Reviewer for this comment which help us to clarify why we only present the SEM images. In fact we also performed some TEM analysis, but the respective results (present below) did not allow us to include additional information to our results regarding the nanoparticles size and dispersity. By comparing the TEM and SEM analysis, it is more perceptible these parameters by SEM results as well as to infer about the NPs shape and morphology, while TEM did not include additional information and for that reason we decide only use the SEM data on the manuscript. In addition, for the determination of size, PDI and charge, in this work it was used DLS. This method was used for measuring sizes since it is based on the frequency of movement, it measures the hydrodynamic radii of the particles, which includes not only the particle itself but also the ionic and solvent layers associated with it in solution under the experimental defined conditions (1), becoming more suitable with the experimental conditions of the present work. It, therefore, models the size from this data. The first parameters obtained from DLS data are the Z-average, the harmonic intensity averaged particle diameter, and the polydispersity index.

(1) Eaton, P.; Quaresma, P.; Soares, C.; Neves, C.; de Almeida, M. P.; Pereira, E.; West, P. A direct comparison of experimental methods to measure dimensions of synthetic nanoparticles. Ultramicroscopy 2017, 182, 179-190.

  1. Figure 6 shows rather poor gel electrophoresis images. Based on the data shown, it is not possible to confirm the band size. First, the ladder needs to be marked to show size information. Second, please consider rerun the gel so that the PCR bands are not located at the bottom of the gel. Especially in panel B, the bands are also cut off.

Answer 2: We thank the Reviewer for this comment which help us to improve our manuscript. In fact the gel electrophoresis images had low information. We rerun the gel and present a new image with more information regarding the range size of the DNA molecular weight marker and the size of E7 transcripts presented in lanes 3, 4 and 5, please see in this revised version Figure 5 in page 15.

Reviewer 2 Report

The manuscript entitled "Modulation of Chitosan-TPP Nanoparticle Properties for 2 Plasmid DNA Vaccines Delivery" describes data on the current problem of finding and improving effective methods of using vaccines against viral infections. To improve the targeted delivery of nucleic acids, the authors used functional nanoparticles, which is consistent with current global trends. The work is written logically, consistently and competently. The illustrations are clear and capacious, the methods are reasonable and reproducible, the detail of their description is impressive. The structure and hierarchy of the article facilitate the perception of the material. The analysis of the material in the review of literary sources is quite deep. The conclusions are substantiated and of scientific value. Based on the fact that in order to create effective methods of gene delivery, many conditions must be met, this work can become the starting point for many studies, for example, for the creation of new vaccines.

The article can be accepted with minor changes.

1) The figures located on pages 8, 12, 14, 15 should be aligned in width.

2) It would be appropriate to add links to new research in the article, for example, in lines 43-45 - 10.1016/j.jbiotec.2020.12.003

3) In Table 4, you can delete uninformative columns CS Concentration and CS Volume. The values in the columns do not differ, there is no need for these columns. The values of 0.02% and 3 ml can be specified in the text or caption to the table.

4) The signature to Figure 7, located on page 15, has been moved to page 16. Perhaps it would be appropriate to combine the figure and the signature on one page.

Author Response

The manuscript entitled "Modulation of Chitosan-TPP Nanoparticle Properties for 2 Plasmid DNA Vaccines Delivery" describes data on the current problem of finding and improving effective methods of using vaccines against viral infections. To improve the targeted delivery of nucleic acids, the authors used functional nanoparticles, which is consistent with current global trends. The work is written logically, consistently and competently. The illustrations are clear and capacious, the methods are reasonable and reproducible, the detail of their description is impressive. The structure and hierarchy of the article facilitate the perception of the material. The analysis of the material in the review of literary sources is quite deep. The conclusions are substantiated and of scientific value. Based on the fact that in order to create effective methods of gene delivery, many conditions must be met, this work can become the starting point for many studies, for example, for the creation of new vaccines. The article can be accepted with minor changes.

Answer: The authors would like to acknowledge the careful revision and valuable and encouraging Reviewer’ comments and the possibility to improve our manuscript, based on your constructive suggestions.

1) The figures located on pages 8, 12, 14, 15 should be aligned in width.

Answer 1: Thank you for this comment which allow us to adapt the figures format, aligning width and make the manuscript format more uniform. Please see figures in page 8, 9, 13 and 15.

2) It would be appropriate to add links to new research in the article, for example, in lines 43-45 - 10.1016/j.jbiotec.2020.12.003

Answer 2: We acknowledge the valuable suggestion of the Reviewer, we included the new research (10.1016/j.jbiotec.2020.12.003) in the line 44, reference 4.

3) In Table 4, you can delete uninformative columns CS Concentration and CS Volume. The values in the columns do not differ, there is no need for these columns. The values of 0.02% and 3 ml can be specified in the text or caption to the table.

Answer 3: The Reviewer is right, there is no need to maintain the columns with CS concentration and CS volume, since the values do not differ. Thus, these columns were deleted and the respective information was specified in the text, please see page11. 

4) The signature to Figure 7, located on page 15, has been moved to page 16. Perhaps it would be appropriate to combine the figure and the signature on one page.

Answer 4: The signature of Figure 7, which in this revised version is Figure 6, was combined to the respective figure in the same page, please see page 15.

Reviewer 3 Report

In this paper, the authors studied the effects CS and TPP volume/concentration on particle size and zeta potential and PDI, and also studied the effect of centrifugation speed on cell viability after transfection with CS-TPP-pDNA polyplexes. Chitosan-TPP nanoparticles are relatively common delivery vectors and are well-studied in literature, therefore the scientific novelty is severely lacking in this work. Furthermore, the variables studied in this paper also lacks scientific importance. For example, centrifugation speed can be easily optimized in a few days or the optimized protocol can be obtained from other papers. The authors need to expand on the biological studies done (e.g. transfection in animal models) or try to experiment with different delivery vectors aside from CS-TPP.

Author Response

In this paper, the authors studied the effects CS and TPP volume/concentration on particle size and zeta potential and PDI, and also studied the effect of centrifugation speed on cell viability after transfection with CS-TPP-pDNA polyplexes. Chitosan-TPP nanoparticles are relatively common delivery vectors and are well-studied in literature, therefore the scientific novelty is severely lacking in this work. Furthermore, the variables studied in this paper also lacks scientific importance. For example, centrifugation speed can be easily optimized in a few days or the optimized protocol can be obtained from other papers. The authors need to expand on the biological studies done (e.g. transfection in animal models) or try to experiment with different delivery vectors aside from CS-TPP.

Answer: We deeply thank the Reviewer for this comment that give us the opportunity to improve our manuscript and clarify the aim of our work. Chitosan is a natural polymer that has been explored in different fields, although with less extension for DNA vaccine delivery, but exhibit several suitable properties for a DNA vaccine delivery such as cationic charge, biocompatibility, biodegradability, mucoadhesiveness and permeability-enhancing properties [1,2]. In this way, and considering the relevance that nucleic acids have gain with the first human approval of a mRNA vaccine against COVID-19, we believe that is an important area to be explored and applied against other virus or diseases such as the cervical cancer induced by human papillomavirus (HPV) infection.    

The mucoadhesion property represents the adherence of a material to a mucosal membrane. The ordinary nanoparticles generally do not adhere to the mucosal surface for more than five hours. To overtake this problem, several polymers have been studied as mucoadhesive agents in particular gene/drug delivery systems [3,4]. Mucoadhesive particles will integrate in the mucus layer via interactions with mucin fibers or electrostatic interactions with the negatively charged mucous layer to improve the entrance of molecules through the mucosal surface. By this way, they prolong the residence time in the mucosal areas, favoring a more effective absorption and controlled release of loaded pharmaceutics [2,4]. In this context, chitosan polymer is the suitable material to develop DNA vaccine delivery nanosystems, when the intention is a needle free administration by mucosal direct application. This vaccination modality can be useful for vaginal mucosa administration of vaccines against HPV, since it is the primary entry route of this virus. Furthermore, vaccines which are delivered directly to the mucosal site can provide a safer and efficient strategy to elicit both systemic and mucosal immunity, comparing to parenteral administration [4,5]. This mucosal vaccination approach also will reduce pain and stresses associated with needle-based injection, eliminate biohazards of needle-disposal and avoid the need of trained medical personnel, all beneficial to overcome the antivaccination movement and increase vaccination rate [5]. In addition, we intend to modify these NPs with specific ligands, such as mannose, to provide a targeted delivery to antigen presenting cells in order to increase the specificity of the vaccine and consequently the intended immune responses. In this way, our research group is already starting these modification studies and after to assure the intended NPS properties will proceed for in vivo studies. We expect to attain interesting results and prepare a future publication but comparing the in vivo performance of different NPs containing mannose ligands (based on synthetic polymers such as PEI (6) and natural polymers in this case chitosan).

 Some information was included in the Introduction section to clarify the relevance of exploring chitosan in DNA vaccine delivery as well as its role as a mucoadhesive polymer, and in conclusions to give a perspective of the future studies to follow, please see pages 2 and 16.

In addition, the authors agree with the Reviewer, and the centrifugation experiments were removed from the manuscript main file and were considered in the supplementary material as cell viability studies. Please see pages 14 to 16 and also the Abstract and conclusion in pages 1 and 16. 

  1. Eusébio, D.; Neves, A.R.; Costa, D.; Biswas, S.; Alves, G.; Cui, Z.; Sousa, Â. Methods to improve the immunogenicity of plasmid DNA vaccines. Drug Discov. Today 2021, doi:https://doi.org/10.1016/j.drudis.2021.06.008.
  2. Netsomboon, K.; Bernkop-Schnürch, A. Mucoadhesive vs. mucopenetrating particulate drug delivery. Eur. J. Pharm. Biopharm. 2016, 98, 76–89, doi:10.1016/j.ejpb.2015.11.003.
  3. Xing, L.; Fan, Y.T.; Zhou, T.J.; Gong, J.H.; Cui, L.H.; Cho, K.H.; Choi, Y.J.; Jiang, H.L.; Cho, C.S. Chemical modification of Chitosan for efficient vaccine delivery. Molecules 2018, 23, doi:10.3390/molecules23020229.
  4. Mohammed, M.A.; Syeda, J.T.M.; Wasan, K.M.; Wasan, E.K. An overview of chitosan nanoparticles and its application in non-parenteral drug delivery. Pharmaceutics 2017, 9, doi:10.3390/pharmaceutics9040053.
  5. Thakkar, S.G.; Warnken, Z.N.; Alzhrani, R.F.; Valdes, S.A.; Aldayel, A.M.; Xu, H.; Williams, R.O.; Cui, Z. Intranasal immunization with aluminum salt-adjuvanted dry powder vaccine. J. Control. Release 2018, 292, 111–118, doi:10.1016/j.jconrel.2018.10.020.
  6. Serra, A.S.; Eusébio, D.; Neves, A.R.; Albuquerque, T.; Bhatt, H.; Biswas, S.; Costa, D.; Sousa, Â. Synthesis and Characteriza-tion of Mannosylated Formulations to Deliver a Minicircle DNA Vaccine. Pharmaceutics 2021, 13, 673, doi:10.3390/pharmaceutics13050673.

Reviewer 4 Report

The chitosan mers are linked by β(1-4) glycosidic bonds not glucosamine residues (line 66). It should be corrected.

What was the molar mass of chitosan?

The SEM analysis in interpreted incorrectly. In the Figure 3 particles with size higher than 500 nm are presented (comparing to scale bar in the right corner). For such analysis a TEM method should be used instead.

Author Response

The authors would like to acknowledge the careful revision and pertinent Reviewers’ comments and the possibility to improve our manuscript, based on their constructive criticism. All the questions are answered below and the recommended modifications were made, being properly highlighted at yellow in the revised manuscript file. 

The chitosan mers are linked by β(1-4) glycosidic bonds not glucosamine residues (line 66). It should be corrected.

Response: We thank the Reviewer’s comment that allow us to improve the manuscript. The sentence was corrected to “by β(1-4) glycosidic bonds”, please see line 68 of page 2.

What was the molar mass of chitosan?

Response: The medical grade chitosan 95/1000 has a molecular weight range between 200 and 500 kDa. I fact, this is an important detail about chitosan that was missing in our manuscript, and we thank the Reviewer for helping us to improve the manuscript. This information was included in line 124, page 3.

The SEM analysis in interpreted incorrectly. In the Figure 3 particles with size higher than 500 nm are presented (comparing to scale bar in the right corner). For such analysis a TEM method should be used instead.

Response: We thank the Reviewer for this comment which help to improve our manuscript. In fact, we performed the SEM and TEM analysis and in the beginning we decided to include only the SEM results as an typical example to show and comment about the NPs shape and morphology. Nevertheless, thinking in the size of NPs, it is more perceptible a in accordance with the DLS results if the TEM analysis are included. In this way, we include in Figure 3 the TEM images. However, the assessment of NPs size, PDI and charge was measured by using DLS and zeta-sizer. This method is more suitable to infer and compare the NPs sizes, since it is based on the frequency of movement, it measures the hydrodynamic radii of the particles, which includes not only the particle itself but also the ionic and solvent layers associated with it in solution under the experimental defined conditions (1), becoming more appropriate with the experimental conditions of the present work. DLS method, therefore, models the size from this data. The first parameters obtained from DLS data are the Z-average, the harmonic intensity averaged particle diameter, and the polydispersity index. Thus, we added the TEM analysis to the manuscript and improve the data analysis and discussion of this figure,  addition and about the data assessment by DLS. Please see pages 4, 5, 13 and 14.

(1) Eaton, P.; Quaresma, P.; Soares, C.; Neves, C.; de Almeida, M. P.; Pereira, E.; West, P. A direct comparison of experimental methods to measure dimensions of synthetic nanoparticles. Ultramicroscopy 2017, 182, 179-190.

Reviewer 5 Report

In this manuscript (polymers-1581788), the authors have modulated the properties of chitosan-TPP nanoparticles (CS-TPP NPs) for plasmid-DNA vaccine delivery. After going through the manuscript, I have found that this study is not much novel because this research opportunity (study) has already been performed as reported elsewhere (Drug delivery, 18(3), 2011, 215-222) and is slightly different. However, this study could be an extension of the published study if the authors provide detailed qualitative in-vitro and in-vivo biological analyses for the same. In my opinion, this manuscript cannot be accepted in its current form, but it can be considered after a major revision in terms of cellular experiments.  Also, the authors should discuss the current study and the published study (Drug delivery, 18(3), 2011, 215-222) comparatively, what has been done or not previously for future research directions.

Author Response

The authors would like to acknowledge the careful revision and pertinent Reviewers’ comments and the possibility to improve our manuscript, based on their constructive criticism. All the questions are answered below and the recommended modifications were made, being properly highlighted at yellow in the revised manuscript file. 

In this manuscript (polymers-1581788), the authors have modulated the properties of chitosan-TPP nanoparticles (CS-TPP NPs) for plasmid-DNA vaccine delivery. After going through the manuscript, I have found that this study is not much novel because this research opportunity (study) has already been performed as reported elsewhere (Drug delivery, 18(3), 2011, 215-222) and is slightly different. However, this study could be an extension of the published study if the authors provide detailed qualitative in-vitro and in-vivo biological analyses for the same. In my opinion, this manuscript cannot be accepted in its current form, but it can be considered after a major revision in terms of cellular experiments.  Also, the authors should discuss the current study and the published study (Drug delivery, 18(3), 2011, 215-222) comparatively, what has been done or not previously for future research directions.

Response: We deeply thank the Reviewer for this comment that give us the opportunity to improve our manuscript and clarify the aim of our work. Chitosan is a natural polymer that has been explored in different fields, although with less extension for DNA vaccine delivery, but exhibit several suitable properties for a DNA vaccine delivery such as cationic charge, biocompatibility, biodegradability, mucoadhesiveness and permeability-enhancing properties. In this way, and considering the relevance that nucleic acids have gain with the first human approval of a mRNA vaccine against COVID-19, we believe that it is an important area to be explored and applied against other virus or diseases such as the cervical cancer induced by human papillomavirus (HPV) infection. Following this idea, we choose to explore chitosan polymer in this work because we consider a suitable material to develop nanosystems to deliver a pDNA vaccine against the HPV directly in the mucosa without resort to needles, due to its properties, especially the mucoadhesion. This vaccination modality can be useful for vaginal mucosa administration of vaccines against HPV, since it is the primary entry route of this virus. Furthermore, vaccines which are delivered directly to the mucosal site can provide a safer and efficient strategy to elicit both systemic and mucosal immunity, comparing to parenteral administration [1,2]. This mucosal vaccination approach also will reduce pain and stresses associated with needle-based injection, eliminate biohazards of needle-disposal and avoid the need of trained medical personnel, all beneficial to overcome the antivaccination movement and increase vaccination rate [2].

In comparison to the work (Drug delivery, 18(3), 2011, 215-222), the novelty of the present study is to produce homogenous chitosan nanoparticles (NPs), without the usage of surfactants or ultrasonication, which can break them, to avoid their aggregation, trying to reach NPs with low polydispersity index (PDI) by only adjusting important parameters. In fact, aggregation of chitosan nanoparticles is a well-known drawback of these systems, which in most of works performed at some years ago, were not controlled or neither the PDI assessed. For instance, Suna ÖzbaÅŸ-Turan & Jülide AkbuÄŸa in their work do not present the PDI, of their CS NPs. The distribution of clusters can be a problem regarding cell internalization, even with small particles. Decreasing the PDI leads to more homogeneous NPs and higher PDI indicates non-uniformity, resulting in broad particle size distribution [3]. Nanoparticles prepared by ionotropic gelation methods are usually recognized by their aggregation/particle fusion directly after preparation or by limited physical/chemical stability when NP suspensions are stored for an extended period of time. Agarwal and colleagues analyzed different chitosan and TPP concentrations to reach the best formulation system, however, they used surfactant tween 80 [0.5% (v/v)], into chitosan solutions in order to prevent particle aggregation [4]. Ozturk and colleagues concluded that using ultra-sonication step into production of nanoparticles provided narrower particle size distribution and smaller PDI. However, the encapsulation efficiency of their systems was very low (bellow 40%), which indicates that ultra-sonication process disrupts the NPs, releasing part of the encapsulated content, and diminishing the encapsulation efficiency rates [5]. Huang and Lapitsky showed that primary 20 - 50 nm NPs were obtained and then aggregated into larger and polydisperse particles obtained at the end of the particle formation process. They suggested that bridging flocculation occurs when a flocculant (in this case TPP) simultaneously binds to two particles and causes aggregation by “bridging” the particles together. This bridging has been proposed as the dominant aggregation mechanism during the formation of CS/TPP nanoparticles [6]. Rampino and colleagues also verified in their study, NPs aggregation occurred while using a high volume of TPP with a low volume of CS. They confirmed that increasing the amount of TPP into the CS solution led to flocculation or particle packing [7]. The same behavior was noticed by Masarudin and colleagues, in their study they showed that above 200 µL of TPP addition, the particle size and PDI increased significantly [8]. In general, all formulations can show good size, encapsulation efficiency and zeta potential. Nonetheless, polydisperse particles/aggregates can be formed in the solution if the PDI is not considered, monitored, and evaluated. Expecting to obtain reasonable results of particle shape and morphology, size distribution, surface charge, and pDNA encapsulation efficiency to ensure cell transfection, gene expression and low cytotoxicity, the issues associated to PDI, aggregation and cluster formation needed to be improved. Thus, in the present work an exhaustive study was performed around optimization of some fundamental parameters, especially the CS and TPP concentrations and volumes added during the ionotropic gelation method, to reach the ideal formulation.

In addition, we intend to modify these NPs with specific ligands, such as mannose, to provide a targeted delivery to antigen presenting cells in order to increase the specificity of the vaccine and consequently the intended immune responses. In this way, our research group is already starting these modification studies and after to assure the intended NPs properties will proceed for in vivo studies. We expect to attain interesting results and prepare a future publication but comparing the in vivo performance of different NPs containing mannose ligands (based on synthetic polymers such as polyethylenimine and natural polymers in this case chitosan).

According all these questions, some information was included in the Introduction section to clarify the relevance of exploring chitosan in DNA vaccine delivery as well as its role as a mucoadhesive polymer; in Results and Discussion to clarify the main focus of the present work around the PDI issue; and in conclusions to give a perspective of the future studies to follow, please see pages 1 – 3, 7, 8, 16 and 17.

References:

[1] Mohammed, M.A.; Syeda, J.T.M.; Wasan, K.M.; Wasan, E.K. An overview of chitosan nanoparticles and its application in non-parenteral drug delivery. Pharmaceutics 2017, 9, doi:10.3390/pharmaceutics9040053.

[2] Thakkar, S.G.; Warnken, Z.N.; Alzhrani, R.F.; Valdes, S.A.; Aldayel, A.M.; Xu, H.; Williams, R.O.; Cui, Z. Intranasal im-munization with aluminum salt-adjuvanted dry powder vaccine. J. Control. Release 2018, 292, 111–118, doi:10.1016/j.jconrel.2018.10.020

[3] Danaei, M.; Dehghankhold, M.; Ataei S.; Hasanzadeh Davarani, F.; Javanmard, R.; Dokhani, A.; Khorasani, S.; Mozafari, M R. Impact of Particle Size and Polydispersity Index on the Clinical Applications of Lipidic Nanocarrier Systems. Pharmaceutics 2018, 10 (2), doi.org/10.3390/pharmaceutics10020057

[4] Agarwal, M.; Agarwal, M. K.; Shrivastav, N.; Pandey, S.; Das, R.; Gaur, P. (2018). Preparation of Chitosan Nanoparticles and their In-vitro Characterization. Int. J. Life Sci. Res. 2018, 4. 1713-1720, doi:10.21276/ijlssr.2018.4.2.17

[5] Ozturk, K.; Arslan, F. B.; Tavukcuoglu, E.; Esendagli, G.; & Calis, S;. (2020). Aggregation of chitosan nanoparticles in cell culture: Reasons and resolutions. Int. J. Pharm. 2020, 578, 119119, doi:10.1016/j.ijpharm.2020.119119

[6] Huang, Y.; Lapitsky, Y.; On the Kinetics of CS/Tripolyphosphate Micro- and Nanogel Aggregation and Their Effects on Particle Polydispersity. J.Colloid Interface Sci. 2017, 486: 27–37, doi.org/https://doi.org/10.1016/j.jcis.2016.09.050.

[7] Rampino, A.; Massimiliano, B.; Paolo, B.; Barbara B.; Attilio, C.; 2013. “CS Nanoparticles: Preparation, Size Evolution and Stability.” International Journal of Pharmaceutics 2013, 455 (1): 219–28, doi.org/https://doi.org/10.1016/j.ijpharm.2013.07.034.

[8] Masarudin, M. J.; Cutts, S. M.; Evison, B. J.; Phillips, D. R.; Pigram, P. J. Factors determining the stability, size distribution, and cellular accumulation of small, monodisperse chitosan nanoparticles as candidate vectors for anticancer drug delivery: application to the passive encapsulation of [14C]-doxorubicin. Nanotechnol. 2015, 67, doi:10.2147/nsa.s91785 

Round 2

Reviewer 1 Report

I have no further comments and recommend the publication of the manuscript in its current form.

Author Response

The authors acknowledge the reviewer's encouraging comments.

Reviewer 3 Report

The previous comments have yet to be addressed. The scientific novelty is still missing. The additional explanation only explains the importance of chitosan as a gene delivery vector and doesn't address the scientific importance and impact their work have as compared to the many other literature that has already been published on the same subject.

Author Response

Response: The authors thank the Reviewer for this comment and tried to better clarify the novelty and difference of their work in relation to previous ones.

The novelty of the present study is to produce homogenous chitosan nanoparticles (NPs), without the usage of surfactants or ultrasonication, which can break them, to avoid their aggregation, trying to reach NPs with low polydispersity index (PDI) by only adjusting important parameters. In fact, aggregation of chitosan nanoparticles is a well-known drawback of these systems, which in most of works performed at some years ago, were not controlled or neither the PDI assessed. For instance, Suna ÖzbaÅŸ-Turan & Jülide AkbuÄŸa in their work do not present the PDI, of their CS NPs. The distribution of clusters can be a problem regarding cell internalization, even with small particles. Decreasing the PDI leads to more homogeneous NPs and higher PDI indicates non-uniformity, resulting in broad particle size distribution [1]. Nanoparticles prepared by ionotropic gelation methods are usually recognized by their aggregation/particle fusion directly after preparation or by limited physical/chemical stability when NP suspensions are stored for an extended period of time. Agarwal and colleagues analyzed different chitosan and TPP concentrations to reach the best formulation system, however, they used surfactant tween 80 [0.5% (v/v)], into chitosan solutions in order to prevent particle aggregation [2]. Ozturk and colleagues concluded that using ultra-sonication step into production of nanoparticles provided narrower particle size distribution and smaller PDI. However, the encapsulation efficiency of their systems was very low (bellow 40%), which indicates that ultra-sonication process disrupts the NPs, releasing part of the encapsulated content, and diminishing the encapsulation efficiency rates [3]. Huang and Lapitsky showed that primary 20 - 50 nm NPs were obtained and then aggregated into larger and polydisperse particles obtained at the end of the particle formation process. They suggested that bridging flocculation occurs when a flocculant (in this case TPP) simultaneously binds to two particles and causes aggregation by “bridging” the particles together. This bridging has been proposed as the dominant aggregation mechanism during the formation of CS/TPP nanoparticles [4]. Rampino and colleagues also verified in their study, NPs aggregation occurred while using a high volume of TPP with a low volume of CS. They confirmed that increasing the amount of TPP into the CS solution led to flocculation or particle packing [5]. The same behavior was noticed by Masarudin and colleagues, in their study they showed that above 200 µL of TPP addition, the particle size and PDI increased significantly [6]. In general, all formulations can show good size, encapsulation efficiency and zeta potential. Nonetheless, polydisperse particles/aggregates can be formed in the solution if the PDI is not considered, monitored, and evaluated. Expecting to obtain reasonable results of particle shape and morphology, size distribution, surface charge, and pDNA encapsulation efficiency to ensure cell transfection, gene expression and low cytotoxicity, the issues associated to PDI, aggregation and cluster formation needed to be improved. Thus, in the present work an exhaustive study was performed around optimization of some fundamental parameters, especially the CS and TPP concentrations and volumes added during the ionotropic gelation method, to reach the ideal formulation.

According to the previous questions already mentioned in the first revision and all these questions, some information was included in the Introduction section to clarify the relevance of exploring chitosan in DNA vaccine delivery as well as its role as a mucoadhesive polymer; in Results and Discussion to clarify the main focus of the present work around the PDI issue; and in conclusions to give a perspective of the future studies to follow, please see pages 1 – 3, 7, 8, 16 and 17.

References:

[1] Danaei, M.; Dehghankhold, M.; Ataei S.; Hasanzadeh Davarani, F.; Javanmard, R.; Dokhani, A.; Khorasani, S.; Mozafari, M R. Impact of Particle Size and Polydispersity Index on the Clinical Applications of Lipidic Nanocarrier Systems. Pharmaceutics 2018, 10 (2), doi.org/10.3390/pharmaceutics10020057

[2] Agarwal, M.; Agarwal, M. K.; Shrivastav, N.; Pandey, S.; Das, R.; Gaur, P. (2018). Preparation of Chitosan Nanoparticles and their In-vitro Characterization. Int. J. Life Sci. Res. 2018, 4. 1713-1720, doi:10.21276/ijlssr.2018.4.2.17

[3] Ozturk, K.; Arslan, F. B.; Tavukcuoglu, E.; Esendagli, G.; & Calis, S;. (2020). Aggregation of chitosan nanoparticles in cell culture: Reasons and resolutions. Int. J. Pharm. 2020, 578, 119119, doi:10.1016/j.ijpharm.2020.119119

[4] Huang, Y.; Lapitsky, Y.; On the Kinetics of CS/Tripolyphosphate Micro- and Nanogel Aggregation and Their Effects on Particle Polydispersity. J.Colloid Interface Sci. 2017, 486: 27–37, doi.org/https://doi.org/10.1016/j.jcis.2016.09.050.

[5] Rampino, A.; Massimiliano, B.; Paolo, B.; Barbara B.; Attilio, C.; 2013. “CS Nanoparticles: Preparation, Size Evolution and Stability.” International Journal of Pharmaceutics 2013, 455 (1): 219–28, doi.org/https://doi.org/10.1016/j.ijpharm.2013.07.034.

[6] Masarudin, M. J.; Cutts, S. M.; Evison, B. J.; Phillips, D. R.; Pigram, P. J. Factors determining the stability, size distribution, and cellular accumulation of small, monodisperse chitosan nanoparticles as candidate vectors for anticancer drug delivery: application to the passive encapsulation of [14C]-doxorubicin. Nanotechnol. 2015, 67, doi:10.2147/nsa.s91785 

Reviewer 4 Report

All suggestions were applied and the article can be published.

Author Response

The authors acknowledge the reviewer's positive comments about the revised version of the manuscript.

Reviewer 5 Report

In my opinion, this manuscript can be accepted now for publication. 

Author Response

(The authors gave the same response as above.)

Round 3

Reviewer 3 Report

The previous comments have yet to be addressed. The scientific novelty is still missing. The additional explanation only explains the importance of chitosan as a gene delivery vector and doesn't address the scientific importance and impact their work have as compared to the many other literature that has already been published on the same subject.

Author Response

The previous comments have yet to be addressed. The scientific novelty is still missing. The additional explanation only explains the importance of chitosan as a gene delivery vector and doesn't address the scientific importance and impact their work have as compared to the many other literature that has already been published on the same subject.

Response: The authors recognize the valuable suggestion of the Reviewer, concerning the impact that in vitro and in vivo studies will bring to this work. But unfortunately, it is not reasonable to perform these studies to include at this stage considering the time window for the resubmission of the current manuscript, and because they will increase the complexity of the work, which at this moment have the focus in the optimization of chitosan/TPP/pDNA nanoparticles (NPs) formulation conditions.

However, ongoing work of our research team, in the pDNA vaccine delivery topic, is focused on chitosan and PEI NPs modification with mannose ligands, to provide a pDNA vaccine targeted delivery to antigen presenting cells in order to increase the specificity of the vaccine and consequently the intended immune responses. The NPs that show better results will be explored in in vivo studies. Hopefully, results from these studies will be reported in a near future.  

The scientific importance and impact of the present manuscript, as well as the difference of this work in relation to previous ones is mainly focused on the production of homogenous chitosan NPs, without the usage of surfactants or ultrasonication, which can break them, to avoid their aggregation, trying to reach NPs with low polydispersity index (PDI) by only adjusting important parameters. In fact, aggregation of chitosan nanoparticles is a well-known drawback of these systems, which in most of works performed at some years ago, were not controlled or neither the PDI assessed. For instance, Suna ÖzbaÅŸ-Turan & Jülide AkbuÄŸa in their work do not present the PDI, of their CS NPs. The distribution of clusters can be a problem regarding cell internalization, even with small particles. Decreasing the PDI leads to more homogeneous NPs and higher PDI indicates non-uniformity, resulting in broad particle size distribution [1]. Nanoparticles prepared by ionotropic gelation methods are usually recognized by their aggregation/particle fusion directly after preparation or by limited physical/chemical stability when NP suspensions are stored for an extended period of time. Agarwal and colleagues analyzed different chitosan and TPP concentrations to reach the best formulation system, however, they used surfactant tween 80 [0.5% (v/v)], into chitosan solutions in order to prevent particle aggregation [2]. Ozturk and colleagues concluded that using ultra-sonication step into production of nanoparticles provided narrower particle size distribution and smaller PDI. However, the encapsulation efficiency of their systems was very low (bellow 40%), which indicates that ultra-sonication process disrupts the NPs, releasing part of the encapsulated content, and diminishing the encapsulation efficiency rates [3]. Huang and Lapitsky showed that primary 20 - 50 nm NPs were obtained and then aggregated into larger and polydisperse particles obtained at the end of the particle formation process. They suggested that bridging flocculation occurs when a flocculant (in this case TPP) simultaneously binds to two particles and causes aggregation by “bridging” the particles together. This bridging has been proposed as the dominant aggregation mechanism during the formation of CS/TPP nanoparticles [4]. Rampino and colleagues also verified in their study, NPs aggregation occurred while using a high volume of TPP with a low volume of CS. They confirmed that increasing the amount of TPP into the CS solution led to flocculation or particle packing [5]. The same behavior was noticed by Masarudin and colleagues, in their study they showed that above 200 µL of TPP addition, the particle size and PDI increased significantly [6]. In general, all formulations can show good size, encapsulation efficiency and zeta potential. Nonetheless, polydisperse particles/aggregates can be formed in the solution if the PDI is not considered, monitored, and evaluated. Expecting to obtain reasonable results of particle shape and morphology, size distribution, surface charge, and pDNA encapsulation efficiency to ensure cell transfection, gene expression and low cytotoxicity, the issues associated to PDI, aggregation and cluster formation needed to be improved. Thus, in the present work an exhaustive study was performed around optimization of some fundamental parameters, especially the CS and TPP concentrations and volumes added during the ionotropic gelation method, to reach the ideal formulation. In this way, we included some information in Introduction, Results and Discussion and conclusions to clarify the main focus of the present work around the PDI issue, please see pages 2, 3, 7, 8, 16 and 17.

We have made an effort to meet, at the possible extent, the requirements pointed out by the reviewer, and we feel the manuscript was considerably improved. In fact, the improvement of the revised manuscript was already recognized by 4 Reviewers that strong recommend its publication the present form. We deeply hope the reviewer can accept this version of our manuscript for publication.

References:

[1] Danaei, M.; Dehghankhold, M.; Ataei S.; Hasanzadeh Davarani, F.; Javanmard, R.; Dokhani, A.; Khorasani, S.; Mozafari, M R. Impact of Particle Size and Polydispersity Index on the Clinical Applications of Lipidic Nanocarrier Systems. Pharmaceutics 2018, 10 (2), doi.org/10.3390/pharmaceutics10020057

[2] Agarwal, M.; Agarwal, M. K.; Shrivastav, N.; Pandey, S.; Das, R.; Gaur, P. (2018). Preparation of Chitosan Nanoparticles and their In-vitro Characterization. Int. J. Life Sci. Res. 2018, 4. 1713-1720, doi:10.21276/ijlssr.2018.4.2.17

[3] Ozturk, K.; Arslan, F. B.; Tavukcuoglu, E.; Esendagli, G.; & Calis, S;. (2020). Aggregation of chitosan nanoparticles in cell culture: Reasons and resolutions. Int. J. Pharm. 2020, 578, 119119, doi:10.1016/j.ijpharm.2020.119119

[4] Huang, Y.; Lapitsky, Y.; On the Kinetics of CS/Tripolyphosphate Micro- and Nanogel Aggregation and Their Effects on Particle Polydispersity. J.Colloid Interface Sci. 2017, 486: 27–37, doi.org/https://doi.org/10.1016/j.jcis.2016.09.050.

[5] Rampino, A.; Massimiliano, B.; Paolo, B.; Barbara B.; Attilio, C.; 2013. “CS Nanoparticles: Preparation, Size Evolution and Stability.” International Journal of Pharmaceutics 2013, 455 (1): 219–28, doi.org/https://doi.org/10.1016/j.ijpharm.2013.07.034.

[6] Masarudin, M. J.; Cutts, S. M.; Evison, B. J.; Phillips, D. R.; Pigram, P. J. Factors determining the stability, size distribution, and cellular accumulation of small, monodisperse chitosan nanoparticles as candidate vectors for anticancer drug delivery: application to the passive encapsulation of [14C]-doxorubicin. Nanotechnol. 2015, 67, doi:10.2147/nsa.s91785